# microRNAs as Promising Biomarkers of Platelet Activity in Antiplatelet Therapy Monitoring

**DOI:** 10.3390/ijms21103477

**Published:** 2020-05-14

**Authors:** Teresa L. Krammer, Manuel Mayr, Matthias Hackl

**Affiliations:** 1TAmiRNA GmbH, 1110 Vienna, Austria; teresa.krammer@tamirna.com; 2King’s British Heart Foundation Centre, King’s College London, London SE5 9NU, UK; manuel.mayr@kcl.ac.uk

**Keywords:** microRNA, platelets, biomarker, antiplatelet therapy, platelet activation, circulating microRNAs, platelet microvesicles, platelet-related miRNA

## Abstract

Given the high morbidity and mortality of cardiovascular diseases (CVDs), novel biomarkers for platelet reactivity are urgently needed. Ischemic events in CVDs are causally linked to platelets, small anucleate cells important for hemostasis. The major side-effect of antiplatelet therapy are life-threatening bleeding events. Current platelet function tests are not sufficient in guiding treatment decisions. Platelets host a broad spectrum of microRNAs (miRNAs) and are a major source of cell-free miRNAs in the blood stream. Platelet-related miRNAs have been suggested as biomarkers of platelet activation and assessment of antiplatelet therapy responsiveness. Platelets release miRNAs upon activation, possibly leading to alterations of plasma miRNA levels in conjunction with CVD or inadequate platelet inhibition. Unlike current platelet function tests, which measure platelet activation ex vivo, signatures of platelet-related miRNAs potentially enable the assessment of in vivo platelet reactivity. Evidence suggests that some miRNAs are responsive to platelet inhibition, making them promising biomarker candidates. In this review, we explain the secretion of miRNAs upon platelet activation and discuss the potential use of platelet-related miRNAs as biomarkers for CVD and antiplatelet therapy monitoring, but also highlight remaining gaps in our knowledge and uncertainties regarding clinical utility. We also elaborate on technical issues and limitations concerning plasma miRNA quantification.

## 1. Introduction

Platelets constitute the second most abundant cell type in blood and are key players in hemostasis and thrombosis. Dysregulation of platelet function is causally connected to ischemic events in patients suffering from cardiovascular diseases (CVDs) (reviewed by [1,2]), the leading cause of death worldwide [3,4]. A plethora of molecules, including microRNAs (miRNAs), has been detected in human platelets [5]. miRNAs are small (~22 nt) noncoding RNAs (ncRNAs) first described in *Caenorhabditis elegans* [6,7], involved in posttranscriptional repression of gene expression by inhibition of translation or degradation of messenger RNAs (mRNAs) [8,9,10,11]. The “seed region” (nucleotides 2–8) enables binding of miRNAs to the 3’ untranslated region (UTR) of mRNAs (reviewed by [12]). A multitude of mRNAs can be repressed by a single miRNA, as the sequence of the binding site can be present in various mRNAs (reviewed by [12]). Individual miRNAs mainly act by fine-tuning protein expression [8]. The detection of extracellular miRNAs in the blood stream—combined with the observation that miRNA expression varies between tissues and is altered in diseases—led to the proposition that circulatory cell-free miRNAs might be suitable biomarkers [13]. In 2011, platelet reactivity was first reported to be linked to the abundance of miRNAs in platelets [14], revealing the potential use of platelet miRNAs as biomarkers of platelet activation. Standard platelet function tests (PFTs) are routinely used to assess platelet activation in CVD patients. The first PFT, light transmission aggregometry (LTA), was developed in the 1960s [15,16]. Since then, many new methods, including point-of-care (POC) assays, have become available (reviewed by [17]). However, many PFTs only allow assessment of one platelet agonist at a time, lack standardization, and require fresh blood. Antiplatelet drugs are successfully used in the prevention of myocardial infarction (MI) or stroke in CVD patients [18]. The benefit of PFT-guided antiplatelet treatment is subject to controversy. Several publications have reported no benefit for patient outcome (reviewed by [17]), whereas other studies and meta-analyses have found PFT guidance in antiplatelet treatment potentially beneficial [19,20,21]. Antiplatelet agents enable potent platelet inhibition but at the expense of a higher risk of hemorrhagic complications. Newer, more potent P2Y12 receptor inhibitors, such as ticagrelor or prasugrel, are associated with an increased risk of bleeding [22,23,24,25]. However, treatment response to pro-drugs such as clopidogrel shows considerable variability [26]. The evaluation of the platelet activation status on antiplatelet therapy might identify patients with inadequate response to platelet inhibition and therefore increased risk for ischemic or thrombotic events [19,20,27]. On the other hand, it could help to identify patients with an excessive response to platelet inhibition and higher risk of bleeding complications. The Testing Responsiveness To Platelet Inhibition On Chronic Antiplatelet Treatment For Acute Coronary Syndromes (TROPICAL ACS) study showed that assessing platelet reactivity during antiplatelet therapy and adjusting the treatment regimen accordingly in a personalized medicine fashion might be beneficial [21]. A reliable miRNA-based biomarker for platelet activation might improve efficient and safe use of antiplatelet drugs and allow the assessment of the in vivo status of platelets. In addition, archived samples could also be analyzed.

## 2. Do Activated Platelets Release miRNAs?

### 2.1. Platelet miRNAs

Over a decade ago, the presence of miRNAs in platelets was reported using samples of patients suffering from polycythemia vera (PV) as well as healthy volunteers [28,29]. However, results from those early studies need to be interpreted with caution because of the potential leukocyte contamination of platelet extracts, which can distort RNA measurements of platelet preparations because of their substantially higher RNA content per cell [30]. In 2009, Landry and colleagues were the first to report on platelet miRNAs using purified, leukocyte-depleted platelet extracts [5]. They identified 219 miRNAs, with miR-142-5p being reported as most abundant, in human platelets by applying locked nucleic acid (LNA) microarray profiling [5]. Follow-up studies varied considerably with regards to the most common miRNAs in platelets. miR-223-3p has been described as the most abundant platelet miRNA in numerous studies [14,31,32,33,34,35,36]. Several years after Landry et al. the first next generation sequencing (NGS) data on platelet miRNAs was published, identifying many more miRNA species. Plé and colleagues described nearly 500 miRNAs present in platelets, with the let-7 family contributing almost half of the total [37]. Bray and colleagues catalogued platelet miRNAs in purified platelets of four healthy volunteers, lifting the number of known platelet miRNAs to approximately 750 [33]. Even though platelets contain substantially less miRNA compared to nucleated cells [38], they contain a diverse array of miRNA species, amounting to ~30% of currently annotated mature human miRNAs (miRBase v22.1) [33].

### 2.2. Origin and Biological Roles of Platelet miRNAs

Since platelets are anucleate, their nucleic acid content has long been presumed static. More detailed examinations of the RNA content of platelets, however, raise doubts whether miRNAs in platelets are actually mere remnants of megakaryocytes. Platelets contain all components (Dicer, Argonaute 2 (Ago2), and TRBP2) necessary for processing precursor miRNA (pre-miRNA) into mature miRNA [5]. As expected in cells lacking a nucleus, Drosha and DGCR8 were not detected [5]. Evidence suggests that while a substantial proportion of the platelet miRNA content originates from megakaryocytes, platelets are also capable of RNA uptake [39]. Megakaryocytes constitute a miniscule part of cells in the bone marrow; investigating their transcriptome is therefore difficult. Also, no scalable primary megakaryocyte cultures are available. In vitro megakaryocyte cultures either use immortalized cell lines, umbilical cord cells, or induced pluripotent stem cells; however, caution must be exercised as these cell cultures do not mirror the in vivo situation of primary megakaryocytes in their bone marrow niche. Various studies using cultured megakaryocytes, however, showed that these cells transcribe a diverse panel of miRNAs, and that the miRNA content of platelets correlates well with the miRNAs found within megakaryocytes [40,41]. Human platelets retain many different mRNAs [42], and selective transfer of mRNAs from megakaryocytes into newly generated platelets has been demonstrated [43]. A similar process regarding the packaging of miRNAs can be envisaged. Aside from data indicating that platelets inherit most of their miRNA content from megakaryocytes, RNA uptake from other cells or the circulation has also been discussed as a contributing factor to miRNA diversity within platelets. The tendency of platelets to endocytose substances from the environment has been established for decades [44]. The capability of platelets to transfer RNA to other cells has been pointed out by several authors [45,46,47]. Clancy and colleagues recently showed that this transfer of RNA is bidirectional by providing evidence of selective mRNA uptake from vascular cells [39]. These findings indicate that platelets might be capable of miRNA uptake from the environment, in addition to inheriting miRNAs from megakaryocytes.

Platelets contain many miRNAs, however, few studies have revealed a connection between individual platelet miRNAs and platelet function. Thus, specific functions of most miRNAs within platelets remain unexplored. One function of platelet miRNAs, the modulation of endogenous protein expression, was reported by Landry and colleagues [5]. They propose that Ago2·miR-223 complexes might be involved in regulating P2Y12 receptor expression in platelets by showing that P2Y12 receptor mRNA is a target of miR-223 [5]. Their claim is further supported by the fact that P2Y12 transcripts were identified in immunoprecipitates with Ago2 [5]. Leierseder and colleagues first assessed the importance of miR-223 in platelet generation and function using miR-223 knockout mice [48]. Surprisingly, the authors report that miR-223 does not impact activation and aggregation of platelets, bleeding time or the development of megakaryocytes [48]. However, the authors used high concentrations of agonists, perhaps missing settled effects of miR-223 [48]. The recovery of platelet numbers following platelet immunodepletion was marginally impaired in mice with global miR-223 knockout, implying a role of miR-223 in thrombopoiesis [48]. Findings from bone marrow transplant experiments suggest that platelet generation is dependent on miR-223 in non-hematopoietic cells [48]. The involvement of miR-223 in the platelet phenotype was subsequently examined by Elgheznawy et al. [49]. Mir-223^y−^ mice exhibited increased platelet aggregate and thrombus formation as well as an increase in thrombus size and occurrence of embolism in the microvasculature [49]. The authors also show that platelets from miR-223-deficient mice display spontaneous aggregate formation in addition to an elevated aggregation response to agonist binding in vitro [49]. Because of these contradictory findings, caution is advised in evaluating the role of miR-223 in the platelet phenotype of mice. Possible compensatory effects in miR-223-deficient mice also need to be considered. Further evidence regarding the influence of platelet miRNAs on platelet reactivity by regulation of protein expression was provided by Kondkar and colleagues [50]. The authors propose that platelet miR-96 is associated with the downregulation of VAMP8/endobrevin, a SNARE protein involved in the secretion of platelet granules, known to be elevated in hyperreactive platelets [50]. Beside miR-223 and miR-96, miR-126 and miR-21 have been implicated in the regulation of platelet function [51,52]. Mice treated with antagomiR-126-3p exhibited a reduced aggregation response upon agonist stimulation [52]. Kaudewitz et al. explain the alteration in platelet aggregation by two mechanisms: (1) miR-126 inhibits expression of ADAM9 (in megakaryocytes), a protease involved in attenuating platelet adhesion to collagen [53], and (2) miR-126 inhibition is linked to decreased P2Y12 receptor expression, essential for platelet activation propagation via released adenosine diphosphate (ADP) in response to strong agonist stimulation [52]. Another group later confirmed that miR-126-3p targets ADAM9, also demonstrating that overexpression of miR-126-3p in megakaryocytes is associated with elevated levels of platelet activation [54]. Platelets from mice treated with antagomiR-21-5p showed an attenuated release of TGF-β1 [51]. Mechanistically, Wiskott-Aldrich syndrome protein, a negative regulator of platelet TGF-β1 secretion, was identified as a direct target of miR-21-5p. miR-21-null mice had lower platelet and leukocyte counts compared with littermate controls but higher megakaryocyte numbers in the bone marrow [51].

### 2.3. Platelets Release Microvesicles Upon Activation

The release of platelet microvesicles (PMVs) in response to stimuli, such as agonist binding or shear forces, is widely accepted (as reviewed by [55]). Around half a century ago, secretion of PMVs was first reported using electron microscopy [56,57] and famously termed “platelet dust” [56]. Almost two decades later, shedding of microvesicles from collagen-activated platelets was visualized with electron microscopy [58]. The premise of activated platelets secreting PMVs has been substantiated by numerous studies [59,60,61] (reviewed by [62]). Aatonen et al. characterized PMVs and reported that the quality and quantity of secreted PMVs differ depending on the activation stimulus [63]. Another study applied cryo-electron microscopy to investigate the size distribution, phenotype, and number of vesicles released by activated platelets [64]. Stimulation of platelet-rich plasma (PRP) resulted in a substantial rise in the number of vesicles compared to unstimulated PRP [64]. The authors report that the majority of vesicles present in stimulated PRP ranges from 50 nm to 1 µm [64]. These findings are in line with previous work demonstrating that stimulated platelets release exosomes (formed from endosomal bodies, <100 nm) as well as microvesicles (formed from plasma membrane, 100 nm–1 µm) [65].

Elevated PMV generation has been described in patients with CVDs compared to healthy individuals (reviewed by [55]). This observation is consistent with high levels of activated platelets in various thrombotic conditions (reviewed by [55]). The fact that activated platelets release PMVs is well established, however, specificities about these vesicles, especially regarding their cargo, remain to be explored. This is due to inherent flaws in the methodology regarding extraction and characterization of extracellular vesicles (EVs) that have only been acknowledged in recent years (reviewed by [66]). Interestingly, it has been suggested that PMV secretion might also occur in quiescent platelets under storage conditions through a mechanism involving integrin signaling [67]. The release of PMVs from resting platelets, mainly because of maturation and apoptosis, has been proposed as a contributing factor to basal PMV levels in healthy individuals [68]. In subsequent experiments, however, no evidence of PMV release from quiescent platelets was found, and the authors concluded that basal levels of PMVs in the blood stream are primarily attributable to activated platelets or release by megakaryocytes [69].

### 2.4. Platelets Secrete miRNAs

In 2008, Hunter and colleagues reported the presence of miRNAs in circulating EVs of healthy individuals [70]. In the same year, blood-derived miRNAs were first described as promising biomarkers of malignant diseases [13]. A substantial fraction of cell-free miRNAs in the blood is platelet-derived, even considering that platelets contain less miRNA per cell than nucleated cells with ongoing transcription [38,71]. The exceptional stability of cell-free miRNAs in the blood has been attributed to the association of miRNAs with proteins (RNA-binding [72] or lipoproteins [73,74]) and the incorporation of miRNAs into EVs [70,75]. The prevailing carrier of extracellular miRNAs, however, remains a matter of debate. Comprehensive studies report that circulatory miRNAs are predominantly transported in complex with Ago2 proteins [72,76]. These results corroborate the notion that the relevance of vesicle-bound miRNAs might have been overestimated as a result of methodological errors and limitations concerning the isolation and characterization of EVs [66]. miRNA species may differ in regard to their carrier: miR-122 appears to be almost exclusively present in protein complexes, whereas other miRNA species, such as let-7a, are predominantly incorporated into vesicles [72]. Regarding miRNA release from activated platelets, it seems probable that miRNAs are mainly secreted within PMVs [75,77]. This hypothesis is supported by the fact that no Argonaute proteins were identified in the protein secretome of activated platelets [78], although this could be due to the low abundance and the lack of sensitivity of the mass spectrometry-based proteomics approach.

A considerable amount of evidence has been gathered in the past decade supporting the hypothesis that activated platelets release miRNAs, possibly resulting in elevated levels of plasma miRNAs (Figure 1).

Alterations of platelet and/or circulatory miRNA levels in conjunction with platelet activation have been investigated by measuring (Table 1):The secretion of miRNAs into buffer plasma;Alterations of the intracellular platelet miRNome;Alterations of miRNA levels in cells after (presumable) uptake of platelet-derived miRNAs;Alterations of platelet-related miRNA levels linked to diseases known to be associated with increased platelet activation (such as CVDs);Alterations of miRNA levels upon antiplatelet therapy.

Evidence regarding the latter two points are presented in the following chapters.

As mentioned above, Nagalla and colleagues related intraplatelet miRNA levels to platelet activation [14]. They identified a number of miRNAs discriminating individuals with hyporeactive or hyperreactive platelet response to agonist stimulation [14]. The differential expression of six platelet miRNAs upon thrombin stimulation was reported shortly thereafter [31]. Another study, investigating miRNA expression in ADP-stimulated platelets and PMVs, suggests directed packaging, as the miRNA composition of PMVs released upon stimulation differed from the original platelet content [75]. The authors also propose that vesicles are the predominant carriers of extracellular platelet miRNAs [75]. In agreement with this finding, Laffont et al. report that secreted miR-223 is predominantly contained within PMVs [77]. Interestingly, they also confirmed the presence of functional Ago2-miR-223 complexes in PMVs [77]. Alterations of the platelet miRNome because of agonist stimulation seem to be independent of the activation pathway [83]. In addition, no substantial stimulus-dependent alterations in the miRNA content of released PMVs were found [36]. Downregulation of platelet miRNAs upon stimulation could indicate the release of these miRNAs. Diminished pre-miRNA maturation could also account for observed downregulation of platelet miRNAs. One feature of hyperreactive platelets is elevated intracellular Ca^2+^ levels, potentially triggering the protease calpain, which in turn deactivates Dicer [49]. Decreased activity of Dicer then leads to reduced pre-miRNA processing [49]. In another study, primary human macrophages were co-cultured with isolated PMVs secreted upon thrombin stimulation [47]. The authors demonstrate that PMVs, containing miR-126-3p, can be internalized by cultured macrophages, resulting in increased levels of the respective miRNA [47]. Recently, Ambrose and colleagues provided further evidence for the secretion of miRNAs upon platelet activation by stimulating washed platelets in buffer [36]. They also showed that ADP is required for the release of miRNAs as secondary mediator [36]. Among the 46 miRNAs consistently present in the supernatants, independent of the activation pathway, miR-223-3p was the most common [36]. Because of methodological concerns, the authors measured miRNAs present in the supernatant without extraction of EVs [36].

## 3. Are Platelet-Related miRNAs Suitable Biomarkers of CVDs?

Circulatory miRNAs are stable in blood [13], and miRNAs within platelets also exhibit robust intraindividual stability, which may allow for single point sampling in case of interindividual comparisons [90]. On the other hand, this also points to the value of serial measurements for assessing treatment response, so each individual can serve as its own control. It is currently unclear whether platelet miRNAs are suitable as means of patient stratification with regard to risk and/or treatment response.

Several aspects make the use of platelet-related miRNAs as biomarker of platelet activation plausible: Platelet activation/function is causally linked to ischemic events in patients suffering from CVDs (reviewed by [1]);Circulatory miRNA levels are altered in CVDs (reviewed by [91]);Platelets are an abundant source of circulatory miRNAs [52,71];miRNAs are released upon platelet activation [36].

As elaborated in the previous chapter, Ambrose et al. [36] report the presence of miRNAs in the supernatant of activated platelets. In the light of biomarker development, it would be compelling to assess whether platelet activation in plasma rather than supernatants would capture alterations in miRNA content. One would expect the miRNA background in plasma to interfere with the measurability of potential changes in platelet miRNA levels. To circumvent this problem, it has been suggested to restrict miRNA measurements to EVs, entailing a more challenging and labor-intensive workflow (reviewed by [92]). Most EVs in plasma and especially in serum are platelet-derived. Assessing whether biomarker candidates are primarily packaged within vesicles or bound to proteins can be beneficial to guide the design of experimental setups. Kaudewitz and colleagues investigated the link between platelet-related miRNA levels in the blood stream and the platelet activation status in the general community (Bruneck cohort) as well as in patients with acute coronary syndrome (ACS) [52]. Plasma levels of miR-223 and miR-126 in ACS patients after 30 days on DAPT showed significant positive correlations with the vasodilator-stimulated phosphoprotein phosphorylation (VASP) assay but not platelet reactivity measurements by LTA [52]. In the general community, plasma levels of miR-223 and miR-126 significantly correlated to platelet proteins (P-selectin, platelet factor 4, and platelet basic protein) [52]. Using a platelet spike-in experiment, the authors show that the respective circulatory miRNAs likely originate from platelets [52]. These findings indicate that platelet-derived miR-223 and miR-126 might be suitable biomarkers of platelet activation [52].

Comorbidities and/or medication have the potential to interfere with plasma miRNA levels and might therefore limit the use of such biomarkers. Diabetes itself has previously been linked to decreased miR-126 levels in the blood stream, suggesting that this comorbidity might require different cut-offs for the use of miRNA-based blood biomarkers [93]. Nevertheless, Olivieri and colleagues report significantly lower plasma miR-126-3p levels in diabetic patients who previously suffered a major adverse cardiac event (MACE) compared to diabetics without such complications [94]. Various platelet-related miRNAs have been suggested as biomarkers of CVDs in the past decade. Zampetaki and colleagues describe an association between CVD in the general community and circulating miR-126 levels [79]. Levels of miR-126 in the general community were only associated with CVD outcomes after adjustment for two other platelet-related miRNAs, miR-223 and miR-197 [79]. None of the miRNAs alone were associated with CVD outcomes [79]. miR-126 and miR-199a have been associated with the risk of thrombotic events in patients suffering from coronary artery disease (CAD) [95]. Interestingly, the authors demonstrate that the effect only becomes significant when assessing vesicle-bound and not total plasma miRNAs [95]. miR-21 has been linked to CVDs in several studies examining heart failure [96], acute myocardial infarction (AMI) [97], and adverse ventricular remodeling after AMI [98]. Cardiac miR-21-5p and plasma miR-320a have been associated with arrhythmogenic cardiomyopathy (ACM) [99,100]. Plasma levels of miR-150-5p have been proposed as biomarker to assess the risk of death in stroke patients [101]. miR-28-3p, another well-known platelet miRNA, has been described as potential biomarker of pulmonary embolism (PE) [102]. Serum miR-223 and miR-197 levels might have predictive value regarding the risk of death in CVDs [103]. Even though a connection between putative platelet-related miRNAs and CVDs has been established over the past years, respective miRNAs need to be rigorously validated and assessed for their capacity to serve as biomarkers. Moreover, examining the cellular origin of circulatory miRNA-based biomarker candidates could provide valuable information that is currently missing in most studies. Identifying an informative miRNA signature encompassing different CVDs and comorbidities is challenging. So far, only few miRNA biomarker candidates have been investigated in detail, leaving further potential candidates to be explored.

### Which Other Cell Types of the Cardiovascular System Express Platelet-Related miRNAs?

The vast majority of platelet miRNAs previously implicated with CVDs are not platelet-specific, but present in various cells of the cardiovascular system. Established platelet miRNAs [33], such as miR-320, miR-223, miR-197, miR-191, miR-150, miR-126, miR-28, miR-24, miR-23a, and miR-21, have also been found in varying abundance in carotid artery tissue [104], endothelial cells (ECs) [105], peripheral blood mononuclear cells (PBMCs) [106], heart tissue [107,108,109], and hematopoietic stem cells (HSCs) [110]. The abundance of platelet-related miRNAs in other cardiovascular cells and their contribution to the pool of circulatory miRNAs requires consideration in the search for platelet reactivity biomarker candidates. For instance, miR-126 expression is confined to ECs and megakaryocytes [52], whereas miR-223 is expressed in platelets and leukocytes [79].

## 4. Are Platelet-Related miRNA Levels Affected by Antiplatelet Therapy?

Excessive platelet activation and thrombus formation in patients suffering from CVDs can be prevented by antiplatelet medication [18]. However, identifying patients with inadequate response using state-of-the-art PFTs remains a challenge [27,111]. A considerable number of patients receiving antiplatelet medication, such as clopidogrel [26] or aspirin (reviewed by [112]) exhibit high levels of platelet activation, termed “high on-treatment platelet reactivity” (HTPR). Patients with HTPR remain at higher risk for the development or recurrence of adverse cardiovascular events, and not all available PFTs reliably identify such patients [111,113,114,115]. A potential mechanism for the resistance to antiplatelet drugs has been suggested recently [116]. La Rosa and co-workers showed that miR-26b-5p is significantly lower in platelets of subjects receiving chronic aspirin treatment [116]. miR-26b-5p is involved in the downregulation of multidrug resistance protein 4 (MRP4) mRNA, encoding an ATP-binding cassette membrane transporter, important for the extrusion of undesired molecules from cells [116]. This finding indicates that miR-26b-5p might be connected to aspirin resistance [116]. A novel biomarker to monitor platelet function during the course of antiplatelet therapy would ideally allow personalized antiplatelet treatment and reliable stratification of patients into “responders” and “non-responders.” Platelet miRNAs are detectable and stable in plasma [13], highly conserved among species [117,118], and enable the in vivo assessment of platelet activation. Platelet-related miRNAs have been suggested as biomarker for the assessment of antiplatelet therapy efficacy by various studies.

### 4.1. Plasma miRNAs Responsive to Antiplatelet Agents

Willeit et al. were the first to examine the responsiveness of platelet-derived miRNAs to platelet inhibition [71]. Escalating doses of aspirin in combination with prasugrel resulted in significantly reduced plasma levels of miRNAs, such as miR-223, miR-191, miR-150, miR-126, in treatment-naïve healthy subjects [71]. This finding was confirmed in a patient cohort with symptomatic carotid atherosclerosis, already receiving aspirin at baseline, after initiation of dual antiplatelet treatment (DAPT) (dipyridamole or clopidogrel) [71]. Another recent study found levels of miR-223, along with miR-223* and miR-197 to be decreased in healthy volunteers after treatment with clopidogrel and ticagrelor [119]. Interestingly, the levels of these miRNAs were elevated in response to experimental endotoxemia despite pharmacological inhibition indicating platelet miRNAs as read-out for P2Y12 receptor-independent platelet activation in endotoxemia [119]. Contrasting previous results, another study found elevated levels of miR-223 under conditions of more potent platelet inhibition, measured by a decrease in platelet aggregability [84]. More potent inhibition of platelet function is achieved by administering newer P2Y12 antagonists, such as ticagrelor or prasugrel, together with aspirin [84]. However, the authors only compared plasma miRNA levels between small groups on DAPT (newer P2Y12 antagonists vs. clopidogrel) without presenting data from healthy volunteers [84]. A recent study investigated alterations in platelet-related miRNAs 24 h after a therapeutic switch from clopidogrel to the more potent platelet inhibitor ticagrelor in ACS patients [85]. Patients receiving the more potent antiplatelet agent had significantly lower plasma levels of miR-223, miR-150, and miR-126 [85]. Platelet reactivity was also significantly reduced after the switch [85]. Administering a loading dose (LD) of ticagrelor further lowered plasma miR-126 levels, suggesting that miR-126 might be especially sensitive to increasing inhibition of platelet function, as this effect was not observed with miR-223 and miR-150 [85]. miR-223, miR-197, and miR-126 have previously been linked to the occurrence of MI in the general population [79]. These findings suggest that a panel of platelet-derived miRNAs measured in plasma might be informative about the effectiveness of platelet inhibition during antiplatelet therapy. While several studies have now confirmed a reduction of platelet-related miRNAs upon initiation of antiplatelet therapy, a recent study reports that circulatory levels of platelet miRNAs do not increase after cessation of prolonged P2Y12 inhibition in CAD patients [87]. Jäger et al. measured known platelet miRNAs previously implicated with platelet function (miR-223, miR-150, miR-126, and miR-21) at baseline and 10, 30, and 180 days following therapy cessation [87]. The authors conclude that platelet-related plasma miRNAs are not suitable as biomarker for platelet activation after discontinuation of P2Y12 inhibition, as they are not influenced by platelet activation but to a certain extent by choice of antiplatelet medication [87].

### 4.2. Platelet miRNAs Responsive to Antiplatelet Agents

Besides circulating miRNAs in plasma or serum, the miRNA content within platelets has also been investigated for potential use as biomarkers for the evaluation of antiplatelet efficacy. Shi and colleagues assessed whether platelet miR-223 and miR-96 levels in Chinese patients suffering from CVD are linked to the responsiveness to antiplatelet therapy [80]. The group reported reduced levels of miR-223 in patients with HTPR to clopidogrel [80]. The authors speculate about the biological implications of this finding, hypothesizing that platelet inhibition might be enhanced in platelets with elevated miR-223 levels due to downregulation of the P2Y12 receptor, as indicated by previous findings [5]. The link between miR-223 and antiplatelet therapy responsiveness has been pointed out by several studies so far [71,84,86], making miR-223 a prime biomarker candidate. In another study, ACS patients were classified as “low responders” and “high responders” to clopidogrel treatment based on the platelet reactivity measured by LTA [86]. Platelet miRNAs were measured at baseline and five days after onset of DAPT (clopidogrel and aspirin) [86]. Levels of miR-223, miR-221, and miR-21 were significantly increased in the platelets of high responders [86]. However, no information regarding heparin use at baseline, which is known to interfere with miRNA quantification by real-time polymerase chain reaction (RT-qPCR), is available [86]. Very recently, Liu and co-workers selected healthy subjects with extremely high or low platelet reactivity, as determined by thromboelastography (TEG) measurements [89]. Healthy volunteers from the high reactivity group exhibited lower levels of miR-223, miR-126 and increased levels of miR-150 [89]. These results were then confirmed in an ACS patient cohort on DAPT with clopidogrel. As in the high platelet reactivity volunteers, patients with HTPR had reduced miR-223 and miR-126 as well as increased miR-150 levels. It is important to emphasize that only extreme cases were analyzed in this study, and the authors do not comment on the potential use of heparin [89].

### 4.3. Plasma miRNAs Responsive to Antiplatelet Treatment in T2DM

Diabetic patients exhibit a marked predisposition for the development of CVDs. However, antiplatelet strategies prove more challenging in diabetic patients because of higher resistance and their propensity for CVD progression. The finding that circulatory miR-126 levels are decreased upon antiplatelet treatment [71] was corroborated in patients with type 2 diabetes mellitus (T2DM) by DeBoer and colleagues [81]. The authors measured miRNAs in platelet-poor plasma (PPP) from 40 diabetics without CVD [81]. Subjects received either a placebo or aspirin for six weeks [81]. After a washout phase, subjects were administered either a placebo or aspirin, depending on the treatment in the first round [81]. Aspirin intake was linked to significantly reduced plasma levels of miR-126 [81]. A shift from platelets to plasma for several miRNAs, including miR-223 and miR-126, suggests that respective miRNAs might be released upon activation of platelets [81]. Lower plasma levels of platelet-derived miRNAs in the context of antiplatelet therapy could indicate reduced secretion due to reduced levels of platelet activation [81].

Similar observations were reported recently by Parker and colleagues, investigating the effects of three antiplatelet agents (clopidogrel, prasugrel, aspirin) on platelet-related plasma miRNA levels and platelet aggregation in patients with T2DM [88]. The authors report reduced levels of miR-223, miR-197, miR-191, and miR-24 in the plasma of patients treated with the P2Y12 inhibitor prasugrel in comparison to aspirin [88]. Interestingly, no significant differences between the different platelet inhibitors were found for miR-126 [88]. T2DM patients receiving aspirin or prasugrel with a history of CVD exhibited decreased levels of plasma miR-197 compared to patients without, rendering this miRNA a possible biomarker candidate for CVD in diabetic patients [88]. In agreement with the findings by Kaudewitz et al. [52], platelet-related plasma miRNA levels were not correlated with ADP- or collagen-induced aggregation [88]. However, circulating miRNA levels correlated with P-selectin expression, indicating that levels of platelet miRNAs in the circulation represent degranulation tendency rather than aggregation response [88]. 

By now, various studies report that certain platelet-related miRNAs, for instance miR-223, are responsive to antiplatelet agents, albeit contradictory results have also been published. Discrepancies between studies might partially be explained by the use of different PFTs. Further investigations into platelet-related miRNA levels in association with clinical outcome in larger collections of patients suffering from CVD would be valuable.

## 5. Which Technical Challenges Need to Be Addressed Regarding the Quantification of Platelet-Related miRNAs in Plasma?

The major challenge for the development of novel platelet miRNA-based biomarkers is the substantial variability introduced by preanalytical and analytical factors, which can exceed the biological variability of interest. Standardized protocols for proper sample handling are essential to improve reproducibility and quality of results. Accurately measuring platelet-derived molecules in the circulation requires minimizing artificial platelet activation and degranulation during plasma sample preparation [120]. Adequate anticoagulation, preferably using citrate-theophylline-adenosine-dipyridamole (CTAD), is imperative to minimize in vitro platelet activation [120]. Storage temperature and time also influence platelets and can distort measurements of platelet-derived molecules [120]. To what extent storage conditions affect miRNA stability remains a matter of debate. Mitchell et al. report that plasma miRNAs (miR-24, miR-16, miR-15b) are stable for 24 h at room temperature [13], whereas Glinge et al. note that miRNAs (miR-29b, miR-21, miR-1) remain stable for 24 h in whole blood but not in plasma [121]. Recently, Faraldi and colleagues proposed that miRNAs vary in their susceptibility to alterations in preanalytical parameters. This could be relevant to many previous studies on preanalytical variability of plasma miRNAs, as often very few miRNAs are assessed [122]. To ensure interstudy comparison, it is paramount to know which compartment of the circulation is assessed. Separating all fractions of the circulation and determining the individual contributions would be ideal, however technically challenging. Extraction of leukocytes and erythrocytes based on density and size is done routinely. Adequate platelet isolation is more demanding, considering how delicately platelets have to be handled to avoid activation. Extraction of microvesicles and exosomes also poses technical challenges that have not been solved so far. miRNA measurements of the cell- and vesicle-free compartment of blood would also be of interest.

### 5.1. Choice of Sample Type: Serum vs. Plasma

Serum represents circulating miRNAs in addition to platelet miRNAs, which are secreted during clotting. This is supported by the finding that some miRNAs are increased in serum, suggesting that coagulation affects levels of circulatory miRNAs [32]. Other miRNAs are reduced in serum, which can be prevented by the use of RNase inhibitors [123]. It is likely that the proteases activated as part of the clotting cascade degrade miRNA carrier proteins, rendering them less stable [124]. PPP on the other hand reflects extracellular miRNA levels, including the in vivo secretome of platelets. The use of PPP, if available, is therefore recommended for the assessment of in vivo platelet activation [124].

### 5.2. Quality of Samples: Cellular Contamination

Cellular contamination of plasma can occur because of incomplete removal of platelets, erythrocytes, or leukocytes and is problematic because of the high RNA content of cells and the low RNA content in plasma (Figure 2). One crucial factor in sample handling is to avoid hemolysis, as lysed erythrocytes release many miRNAs, altering the measured levels of respective miRNAs [125]. Kirschner and co-workers demonstrated that even mild hemolysis without visually recognizable red discoloration of samples can affect miRNA levels [126]. Residual platelets are easily activated in vitro, for example by mechanic stimuli [83], distorting the assessment of circulatory platelet-derived miRNAs. The importance of an adequate centrifugation protocol to remove cellular contamination has been pointed out by several studies [127,128,129,130]. Basso et al. recommend double centrifugation to remove residual platelets [130]. This is supported by Binderup et al. reporting significantly increased miRNA levels in standard plasma (single centrifugation) compared to PPP (double centrifugation) [128]. The same group later investigated the effect of prolonged single centrifugation compared to double centrifugation on plasma miRNA levels (miR-126, miR-92a, miR-16) and found no significant differences [131]. However, as expected, standard plasma (single centrifugation) and PPP were poorly correlated [131]. Mitchell et al. also pointed out that many studies investigating plasma miRNAs in the context of diseases have not addressed platelet contamination properly [129]. Residual platelets in archived plasma pose a common problem, which cannot be solved by an additional centrifugation step after a freeze/thaw cycle [129]. Freezing/thawing of residual platelets in standard plasma itself is linked to generation of PMVs and miRNA release, rendering additional centrifugation steps after thawing ineffective [129]. Interestingly, platelet contamination seems to be reduced in citrated plasma in comparison to ethylenediaminetetraacetic acid (EDTA) plasma [129]. This finding might be due to increased platelet aggregation in blood anticoagulated with citrate, allowing more efficient removal of platelets through centrifugation, as previously reported [132]. EDTA has also been shown to be a potent activator of vesicle release. The findings by Mitchell et al. are contradicted by an earlier study concluding that post-thaw centrifugation is effective in removing platelet contamination from standard plasma [133]. Cheng et al. measured miRNAs, microvesicles, and platelet count in archived samples either subjected to an additional post-thaw centrifugation step or not [133]. However, the authors did not compare fresh PPP to post-thaw PPP, and therefore did not contemplate that the freeze/thaw process itself causes vesicle and miRNA release, as established by Mitchell and colleagues [129].

### 5.3. Limitations Due to Heparin Use

Patients undergoing coronary interventions for CVD routinely receive a bolus of heparin to prevent coagulation. Heparin is, however, well-known for interfering with polymerase chain reactions (PCRs). If samples from heparin-treated patients were not eligible for miRNA analysis, the use of miRNA-based in vitro diagnostics (IVDs) would be limited. The effect of heparin in plasma was reported to strongly affect exogenous RNA controls and to a lesser extent endogenous miRNAs [134]. The effect was linked to the half-life of heparin in the circulation and not detectable anymore 6 h after the bolus injection [134]. Recently, Schulte and colleagues showed that heparinase treatment of respective samples is essential for analysis of exogenous as well as endogenous RNAs and offers a solution to include samples with heparin in RNA measurements [135].

## 6. Conclusions

We discussed four of the most pressing questions concerning the development of a novel miRNA-based biomarker for platelet activation: Do activated platelets release miRNAs?Are platelet-related miRNAs suitable biomarkers of CVDs?Are platelet-related miRNA levels affected by antiplatelet therapy?What technical challenges need to be addressed regarding the quantification of platelet-related miRNAs in plasma?

Multiple issues regarding the utility of platelet-related miRNAs as biomarkers are needed to be addressed in future experiments and studies. As of now, there is no consensus on a standardized protocol for sample handling to minimize preanalytical variability. The impact of comorbidities and medication intake is largely unknown. In addition, assessing the cellular origin of circulating miRNAs of interest as well as the contribution of other cells/tissues to the extracellular pool of miRNAs would be useful. Assessing the kinetics of platelet miRNA release and clearance in response to activation could provide valuable information as well. To our knowledge, it has never been investigated in which subcellular compartment miRNAs are localized within platelets. The significance of the miRNA background in plasma, possibly impeding the detection of moderate changes in miRNA levels, remains to be elucidated. Restricting the analysis of circulating miRNAs to subpopulations of plasma, such as EVs or protein-bound miRNAs, might be beneficial.

Even though questions remain to be answered, platelet-related miRNAs are promising biomarker candidates. Platelets host a diverse array of miRNA species, some of which are potentially involved in regulating platelet reactivity. PMVs are released upon platelet activation, and a few studies have established that stimulated platelets release miRNAs, possibly within PMVs. Platelets substantially contribute to the pool of cell-free miRNAs in the blood. Aberrant activation of platelets is causally associated with ischemic events in CVDs, and platelet-related miRNA levels are altered in patients suffering from CVDs. Standard PFTs, routinely applied in clinics and research, have certain disadvantages. The use of PFTs to reliably guide treatment decisions is controversial. Right now, the clear benefits of platelet inhibitors come at the expense of a higher risk of bleeding complications. Proper assessment of patients on antiplatelet therapy may identify “low responders” or patients at risk of potentially life-threatening cardiovascular events. A novel miRNA biomarker for platelet activation would not only allow analysis of archived samples, but also provide a read-out that may be closely related to in vivo platelet function. In general, a biomarker should be measurable in a clinical setting, add new information/improve standard tests, and guide medical decisions [136].

A dependable miRNA-based biomarker of platelet activation would need to: Avoid confounding by preanalytical and analytical variation;Show a strong correlation between the candidate miRNA and platelet activation;Guide treatment decisions to antiplatelet agents.

So far, platelet-related miRNAs do not fulfill these criteria yet, however, a substantial body of research has demonstrated high potential, warranting further examination of miRNA biomarker candidates of platelet activation.

## Figures and Tables

**Figure 1 ijms-21-03477-f001:**
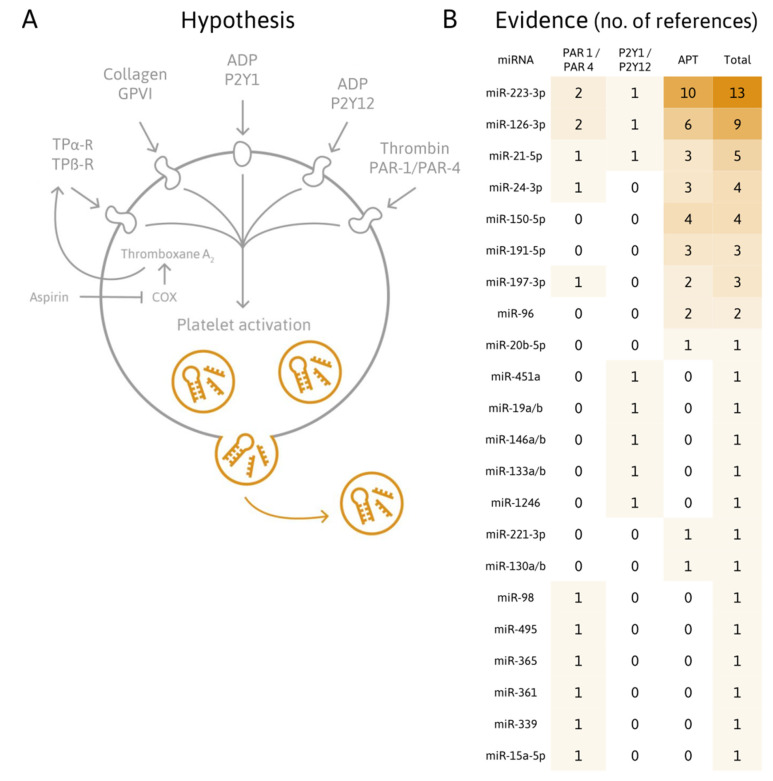
Do activated platelets secrete miRNAs? (**A**) Platelets release miRNAs upon activation by various agonists. Therefore, changes in plasma concentrations of platelet-derived miRNAs might inform on platelet activation and function and have clinical utility for monitoring antiplatelet therapy and cardiovascular diseases (CVDs). (**B**). Heatmap summarizing the number of studies cited in this review, which attempted to assess miRNAs in the context of platelet stimulation (through PAR 1/4 and P2Y1/P2Y12 receptors) and antiplatelet therapy, thus, providing evidence in support of this hypothesis.

**Figure 2 ijms-21-03477-f002:**
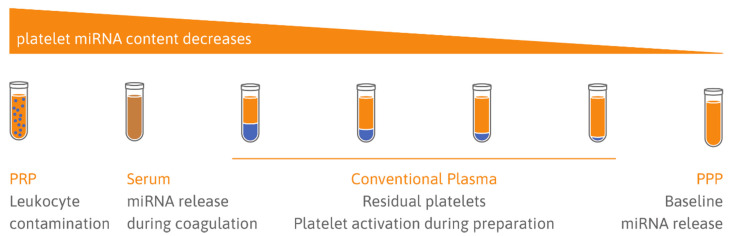
Platelet miRNA content in different blood samples. In platelet-rich plasma (PRP), residual leukocytes can distort miRNA measurements. Serum reflects platelet activation in conjunction with possible miRNA release as well as potentially increased degradation of miRNAs. Conventional plasma reflects the circulatory cell-free miRNA content and platelet miRNAs, depending on residual platelet count and artificial platelet activation during sample preparation. PPP is probably best suited to study the in vivo platelet secretome (adapted from [124]).

**Table 1 ijms-21-03477-t001:** Studies investigating the association between platelet and/or circulatory miRNA levels and platelet activation.

Evidence	Reference	Year	miRNAs of Interest	Agonist/Condition/Treatment	PFT	Origin of Samples	Method
B	[14]	2011	74 miRNAs differentially expressed	Hyperreactive vs. hyporeactive platelets	Maximal aggregation response to ADP and epinephrine	19 healthy subjects	S
B	[31]	2011	miR-15a miR-98 miR-339 miR-361 miR-365 miR-495	Thrombin	P-selectin	4 stimulated, 6 resting platelet samples from healthy subjects	S
A + B + D	[75]	2012	miR-1246 miR-451 miR-223 miR-146 miR-133 miR-126 miR-21 miR-19	ADP and patients with stable CAD vs. patients with ACS	-	Extracted platelets, 5 patients with stable CAD, 5 patients with ACS	T
A + B + D	[79]	2012	miR-223 miR-197 miR-126 miR-24 miR-21	Thrombin and healthy subjects: limb ischemia-reperfusion	-	820 subjects from general population (Bruneck cohort), 11 healthy subjects, extracted platelets and PMVs	S + T
A + B D + E	[71]	2013	miR-223 miR-197 miR-191 miR-150 miR-126 miR-24 miR-21 miR-20b	Healthy subjects: dose-escalation of ASA combined with prasugrel Patients: ASA at baseline, addition of dipyridamole or clopidogrel	Verify Now, LTA, formation of thromboxane B_2_	Platelets, PMVs, serum, PRP, PPP from 3 healthy subjects, serum and PPP from 19 T2DM patients, 9 healthy subjects, 33 patients with symptomatic carotid atherosclerosis	S + T
B + E	[80]	2013	miR-223 miR-96	Clopidogrel + ASA	VASP assay, LTA	33 non-diabetic CVD patients	T
A + C	[77]	2013	miR-223	Thrombin and co-incubation of HUVECs with released PMVs	P-selectin	Extracted platelets, HUVEC	T
A + D + E	[81]	2013	miR-126	PRP stimulated with AA in presence/absence of aspirin and patients: one period (6 weeks) placebo, one period ASA	P-selectin	4 healthy subjects, 40 T2DM patients without CVD	T
E	[82]	2014	miR-223	DAPT with clopidogrel	VASP assay	62 ACS patients	T
B	[83]	2015	~50 miRNAs differentially expressed	ADP, collagen, TRAP	LTA	15 healthy subjects	S
D	[49]	2015	miR-223	T2DM	Clot retraction & platelet adhesion and spreading assay	22 patients with T2DM, 22 healthy subjects	T
E	[84]	2015	miR-223	DAPT with clopidogrel or prasugrel or ticagrelor	MEA	21 patients with ACS	T
D + E	[52]	2016	miR-223 miR-191 miR-126 miR-24	Healthy subjects: DAPT with prasugrel and patients: ASA only or DAPT with clopidogrel, prasugrel, or ticagrelor	VASP assay, LTA, Verify Now	669 subjects from general population (Bruneck cohort), 125 ACS patients, additional ACS patients (*n* = 8/group) and healthy subjects (*n* = 6) to assess impact of antiplatelet treatment	S
C	[47]	2016	miR-126	Thrombin stimulation of platelets, co-incubation of PMVs with primary human macrophages	-	Healthy subjects	S + T
E	[85]	2016	miR-223 miR-150 miR-126 miR-96	Switch from DAPT with clopidogrel to ticagrelor “No load”: ASA + ticagrelor without LD“Load”: ticagrelor LD, then ticagrelor MD + ASA	MEA	16 ACS patients (8 “no load” group, 8 “load” group)	T
E	[86]	2017	miR-223 miR-221 miR-21	DAPT with clopidogrel	LTA	272 subjects included; 21 “high responders”, 18 “low responders”	T
A	[36]	2018	46 miRNAs consistently secreted	CRP-XL, PAR1-AP, PAR4-AP, ADP	-	4 healthy subjects	S
E	[87]	2019	miR-223 miR-150 miR-126 miR-21	Cessation of DAPT with clopidogrel or prasugrel or ticagrelor	MEA	62 CAD patients	T
E	[88]	2020	miR-223 miR-197 miR-191 miR-24	ASA + one period (28 days) clopidogrel, one period prasugrel	LTA, P-selectin	56 T2DM patients	T
E	[89]	2020	miR-223 miR-150 miR-130 miR-126	Patients: ASA + clopidogrel	TEG	214 healthy subjects, 430 ACS patients	T

(A) The secretion of miRNAs into buffer and plasma, (B) alterations of the intracellular platelet miRNome, (C) alterations of miRNA levels in cells after (presumable) uptake of platelet-derived miRNAs, (D) alterations in miRNA levels linked to diseases known to be associated with increased platelet activation (such as CVDs), and (E) alterations of miRNA levels upon antiplatelet therapy. (S = screening (NGS, microarray, high-throughput qPCR), T = targeted approach (individual qPCRs)). AA = arachidonic acid, ACS = acute coronary syndrome, ADP = adenosine diphosphate, ASA = acetylsalicylic acid (aspirin), CAD = coronary artery disease, CRP-XL = crosslinked collagen-related peptide, DAPT = dual antiplatelet therapy, HUVEC= human umbilical vein endothelial cell, LD = loading dose, LTA = light transmission aggregometry, MD = maintenance dose, MEA = multiple electrode aggregometry, PAR2-AP = protease-activated receptor-2 activating peptide, PAR4-AP = protease-activated receptor-4 activating peptide, PMV = platelet microvesicle, PPP = platelet-poor plasma, PRP = platelet-rich plasma, T2DM = type 2 diabetes mellitus, TEG = thromboelastography, TRAP = thrombin receptor activating peptide, VASP = vasodilator-stimulated phosphoprotein phosphorylation.

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
