# Peer review of "microRNAs as Promising Biomarkers of Platelet Activity in Antiplatelet Therapy Monitoring"

_ijms, 2020, doi:10.3390/ijms21103477_

Round 1

Reviewer 1 Report

The manuscript entitled “microRNAs as biomarkers of platelet activity in antiplatelet therapy monitoring” by Teresa L. Krammer et al., accurately outlines the complexities of microRNAs as biomarkers for platelet activities. Overall, the authors have been trying to prepare the conceptional manuscript. There are a few nit-picky comments. Since authors demonstrated the both megakaryocyte and platelets, it is possible to mention that platelet can also retain messenger RNA (with related reference(s)). It will be more helpful for the readers. One more comment is that when mentioning miRNAs, it is definitely better to describe if they are -3p or -5p, as many as possible (for example, miR-126 and miR-126-3p).

Reviewer 2 Report

The review by Krammer et al focusses on platelet-miRNAs and their potential value as biomarkers of platelet activity, which refers to an actual and still unresolved issue. The review is well written and, the subject is thoroughly and extensively revised and well referenced. Cited articles are up to date and the few reviews included are embedded in the text for further reading.

Some suggestions regarding the manuscript are:

The abstract section should summarize more accurately the still unresolved situation regarding the use of platelet miRNA as potential biomarkers as it is discussed throughout the manuscript.

Title needs to be revised. The unresolved situation regarding the usefulness of platelet-miRNAs as biomarkers should be highlighted. The potential use of miRNAs as biomarkers of antiplatelet therapy is only one of the different still open questions discussed in the manuscript.

Chapter 2.3 is beyond the scope of the manuscript. Authors should consider removing it. There is extensive literature regarding platelet microvesicles. Here, as presented, it difficults reading of the manuscript.

Chapter 3.1 needs to be addressed in more detail. Circulating miRNAs directly depend on a complex interplay between platelets, immune cells, and erythrocytes.  This should be highlighted in the manuscript and more extensively documented in order to give a more accurate picture of the value of circulating miRNAS as biomarkers of platelet activity.   

A figure illustrating the main message given in the manuscript would be appreciated.

Minor comments

Text needs to be revised for typing errors

i.e line 454. EDTA is written ETDA;  “in vivo” or “in vivo”.

Page 1, line 41.   Please revise the expression  “Several Years later”  which was the reference point?
